# PIE-scope, integrated cryo-correlative light and FIB/SEM microscopy

**Sergey Gorelick[1,2], Genevieve Buckley[1,2], Gediminas Gervinskas[3], Travis K Johnson[4], Ava Handley[5], Monica Pia Caggiano[1,2], James C Whisstock[1,2,6,7], Roger Pocock[5], Alex de Marco[1,2,6]\***

[1]ARC Centre of Excellence in Advanced Molecular Imaging, Monash University, Clayton, Australia; [2]Department of Biochemistry and Molecular Biology, Biomedicine Discovery Institute, Monash University, Clayton, Australia; [3]Ramaciotti Center for Cryo-Electron Microscopy, Monash University, Clayton, Australia; [4]School of Biological Sciences, Monash University, Clayton, Australia; [5]Department of Anatomy and Developmental Biology, Biomedicine Discovery Institute, Monash University, Clayton, Australia; [6]University of Warwick, Coventry, United Kingdom; [7]EMBL Australia, Clayton, Australia

**Abstract** Cryo-electron tomography (cryo-ET) is emerging as a revolutionary method for resolving the structure of macromolecular complexes in situ. However, sample preparation for in situ Cryo-ET is labour-intensive and can require both cryo-lamella preparation through cryo-focused ion beam (FIB) milling and correlative light microscopy to ensure that the event of interest is present in the lamella. Here, we present an integrated cryo-FIB and light microscope setup called the Photon Ion Electron microscope (PIE-scope) that enables direct and rapid isolation of cellular regions containing protein complexes of interest. Specifically, we demonstrate the versatility of PIE-scope by preparing targeted cryo-lamellae from subcellular compartments of neurons from transgenic *Caenorhabditis elegans* and *Drosophila melanogaster* expressing fluorescent proteins. We designed PIE-scope to enable retrofitting of existing microscopes, which will increase the throughput and accuracy on projects requiring correlative microscopy to target protein complexes. This new approach will make cryo-correlative workflow safer and more accessible.
DOI: https://doi.org/10.7554/eLife.45919.001

**\*For correspondence:**
alex.demarco@monash.edu

**Competing interests:** The authors declare that no competing interests exist.

## Introduction

Cryo-electron tomography (cryo-ET) is currently the principal method for investigating the structure of proteins and protein complexes directly in their native environment (*Beck and Baumeister, 2016*). Here, a cellular sample is vitrified by plunge or high-pressure freezing to instantaneously cease cell activity and prevent crystallisation of water in the sample (*Dubochet et al., 1988*) (*Lucić et al., 2005*). Despite the high-resolution potential of cryo-ET, there are two major challenges that limit the applicability of this technique. First, it is not possible to image a cell in regions thicker than 500 nm using a conventional 300 keV transmission electron microscope (*Frank, 1996*). Second, the cellular environment is extremely heterogeneous and it is impossible to unequivocally identify every visible density (*Förster et al., 2010*). To overcome the issue of cell thickness, the most common and successful approach is to use a cryo-focused ion beam microscope (FIB) to thin the sample and produce flat electron-transparent lamellas of approximately 300 nm thick (*Marko et al., 2006*; *Marko et al., 2007*) (*Hsieh et al., 2014*; *Rigort et al., 2010*). Thus, in a single lamella, only a portion of the cell is retained for cryo-ET but it remains unknown as to whether the protein/complex of interest is present. Considering that every step from FIB milling to cryo-ET imaging is time-

consuming, it is crucial to accurately target the FIB milling procedure to the region on interest. The solution to this has been correlative light and electron microscopy (CLEM) (*Arnold et al., 2016*; *Sartori et al., 2007*), which also offers the opportunity to validate the interpretation of the cryo-ET data by matching the fluorescence data with tomographic densities.

The most complete cryo-CLEM workflow links light microscopy (live cell imaging and cryo-LM), cryo-FIB and cryo-ET (*Figure 1*). The goal, in this case, is to image a protein of interest within a living specimen at the highest possible/required resolution (this step is optional but it can be crucial when

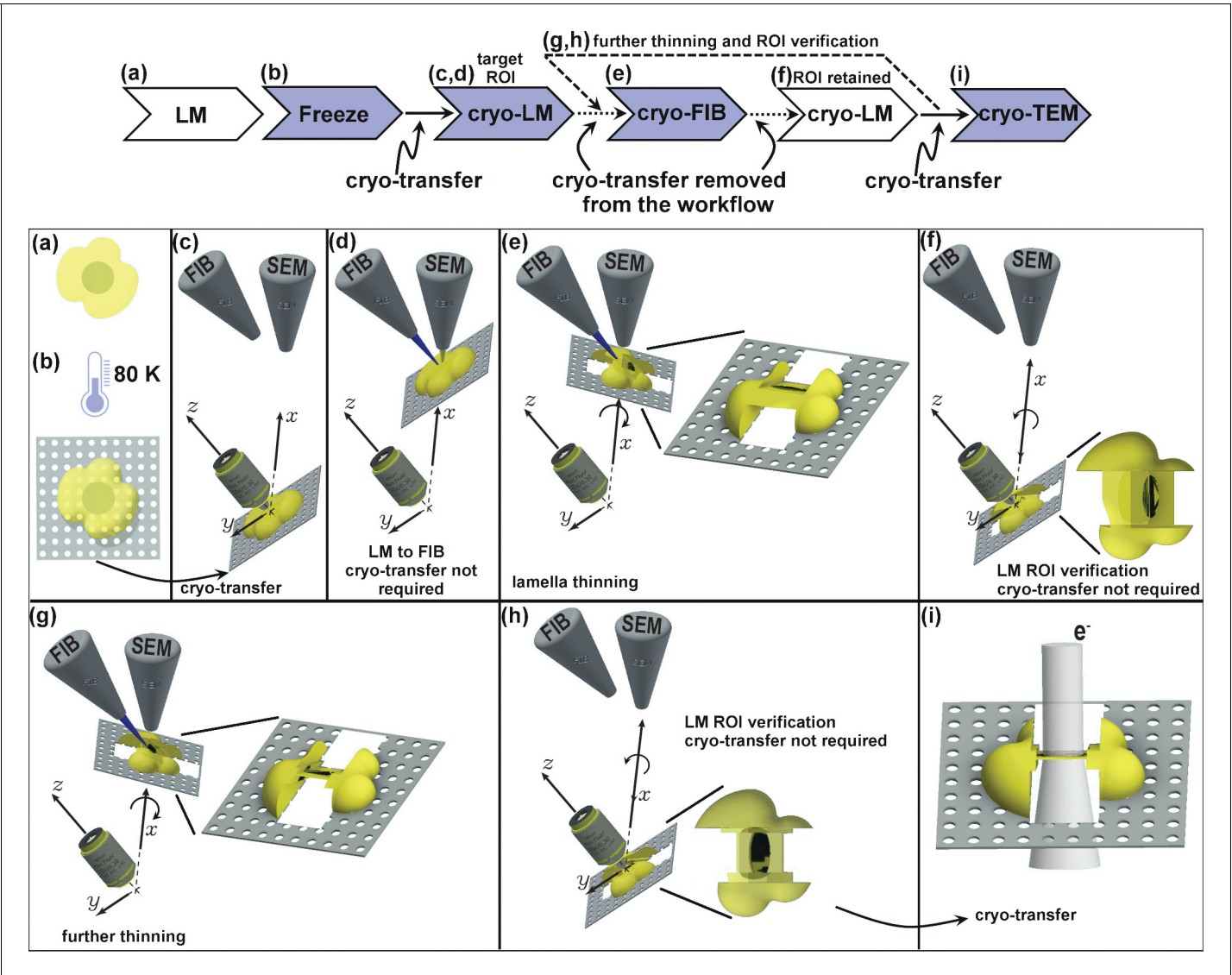

**Figure 1.** Schematic presentation of the workflow. (a) Live cell imaging. (b) Plunge or high-pressure freezing of the sample for cryo-preservation (note, cells can be either cultured on the grid in (a) or in case of suspension be added onto the grid at this stage). (c) Cryo-light fluorescence microscopy in the PIE-scope to identify regions of interest. (d) Simple lateral translation of the sample brings the selected region of interest under the FIB/SEM columns. (e) Once the region of interest is under the FIB, a lamella can be thinned down. The angle between the prepared lamella to the grid plane can be controlled by the stage tilt. (f) Simple translation and tilt of the prepared lamella inside the system allow verification of successful targeting of the region of interest with light microscopy. (g,h) An optional repeat of (e) and (f) for further thinning of the lamella. Since two cryo-transfers are required, the system allows rapid and multiple intermediate verifications of targeting with optical and scanning electron microscopes. The x,y,z-axes shown in c-h correspond to the stage axes of the FIB/SEM. (i) Once the region of interest is successfully targeted and optically verified, the lamella is cryo-transferred into a TEM for high-resolution sample analysis. Distances and dimensions throughout the figure have been adapted for illustrative purposes and are not representative.

DOI: https://doi.org/10.7554/eLife.45919.002

working on time-sensitive cellular processes). Then, when appropriate, the sample is vitrified though plunge freezing (or high-pressure freezing). At this stage, the sample must be transferred under cryogenic conditions into a cryo-LM (*Sartori et al., 2007*; *Schorb and Briggs, 2014*; *van Driel et al., 2009*) and imaged to obtain a still after cryo-fixation to locate all the regions that will be targeted for cryo-FIB milling. Cryo-LM on the vitrified sample is required (even if fluorescence images have been acquired on the living sample) to ensure that the images recorded during the living specimen experiment can be related to the cryo-LM image as effects due to dynamic cell process, blotting and plunge-freezing could cause the target structure to move position. The sample is then transferred (again under cryogenic conditions) into the cryo-FIB. Once it has been milled, the specimen needs to be either transferred to the cryo-TEM for imaging or back to the cryo-LM to confirm that the FIB milling successfully isolated the region housing the protein of interest (*Arnold et al., 2016*).

This workflow requires handling of samples across multiple microscopes (*Arnold et al., 2016*; *Rigort et al., 2010*). Each transfer must be performed under cryogenic conditions and in a humidity-free environment in order to avoid the formation or deposition of crystalline ice. Also, handling a cryo-EM grid on which multiple lamellas have been prepared is an extremely delicate process. Even though users do not handle a bare grid as in the formative days of this field (*Marko et al., 2007*; *Rigort et al., 2010*), but rather a cartridge (which increases the sample stiffness), every manual handling step risks thawing, damaging or contaminating the sample and lowers the overall success of the process (*Hsieh et al., 2014*). This also means that users are unlikely to confirm whether their lamella preparation was successful via cryo-LM before acquiring the data with a transmission electron microscope (TEM), which may further impact workflow success.

All the risks associated with cryo-transfers described above can be mitigated using an integrated approach, where a light microscope is integrated into the vacuum chamber of the electron microscope. Such an approach has been successfully implemented for CLEM where the introduction of a light microscope objective in a TEM column permitted switching between TEM and LM imaging, known as the iLEM (*Agronskaia et al., 2008*). The iLEM has demonstrated its value on both room temperature and cryo-samples (*Faas et al., 2013*), making navigation seamless and accelerating the data collection process by eliminating the need to image areas not representing the region of interest. The major limitations that iLEM suffers from are that high-resolution LM imaging is impossible, because of the tight geometry of the TEM column, and the need to remove the cryo-box usually protecting the sample, which can lead to long-term contamination. Also, for the specific requirement of targeting the lamella preparation, having a light microscope embedded in the cryo-TEM will not be beneficial.

Here, we report the design and functionality of an integrated cryo-LM and cryo-FIB/SEM which we call the Photon Ion Electron Microscope, or PIE-scope. We demonstrate how the PIE-scope can be used to accurately guide cryo-FIB milling to target cells and tissues using a variety of specimens. We also present an example of software integration to control both the cryo-FIB and the cryo-LM.

## Results and discussion

### The PIE-scope setup

To ensure that existing FIB/SEM systems in the field can be retrofitted with the solution proposed here, we chose a design that would be applicable to as many systems as possible regardless of the SEM working distance and column shape. For this reason, the LM does not image the sample at the coincidence point but 49 mm away from it along the X axis (*Figure 2*). Although we developed it to fit on a standard ThermoFisher FIB/SEM chamber, with minor modifications it can be adapted to other systems.

The position of the objective relative to the FIB/SEM offers multiple advantages: (i) switching between the LM and FIB/SEM imaging can be performed with a single axis translation (*Figure 1*); (ii) no rotation is required - only tilt might be needed in some designs (important for cryo-stages where rotation is limited); (iii) if the geometry of the sample holder on the stage allows, objectives with high numerical aperture can be used; (iv) focussing with the LM does not require the sample to be moved.

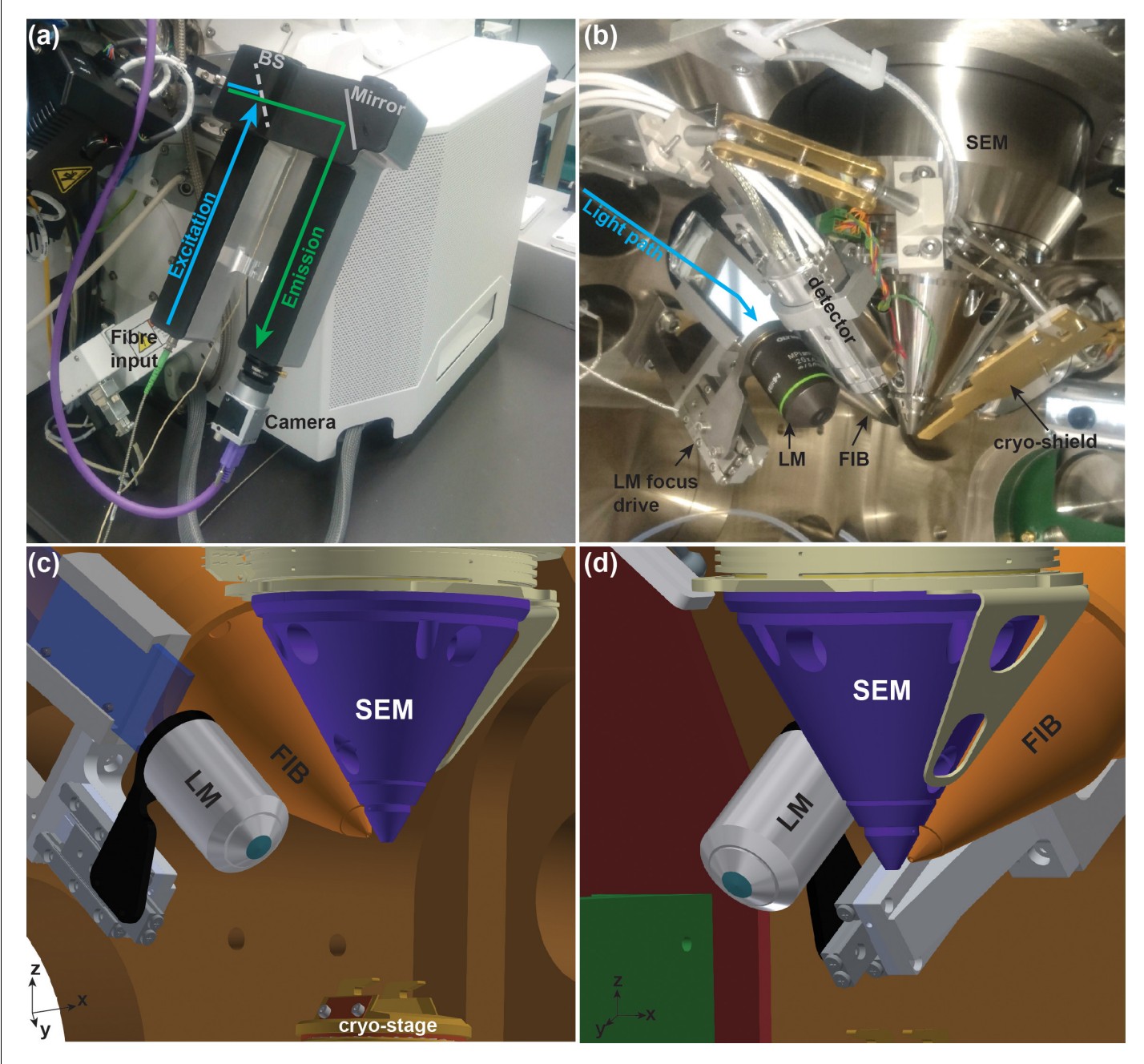

**Figure 2.** Integrated cryo-FIB and light microscope setup (PIE-scope). (a) Outside view of the PIE-scope. The atmospheric (external) retrofitted component consisting of one excitation and one emission arms. The camera at the end of the emission arm can be replaced. The excitation arm has been designed to fit an FC/APC fibre-end with a 6 μm core and a NA = 0.06. BS represents the position of the dichroic beam-splitter. (b) A view of the FIB/SEM chamber containing showing the in-vacuum section of the PIE-scope. An objective is mounted on a high-precision motorized stage for sample focusing (LM focus drive). The light from the external arm (cyan arrow) is delivered through a glass flange and directed into the objective by a mirror. (c, d) CAD renderings of the in-vacuum section of the PIE-scope. These renderings are a simplified representation of the chamber and do not include all components visible in (b), but the renderings clarify the geometry of the critical components of PIE-scope. The XYZ axes at the bottom left of panels c and d are aligned with the stage axes.

DOI: https://doi.org/10.7554/eLife.45919.003

When designing LM assembly, we considered that if not in use, the objective should be retracted to prevent collision with the stage. Furthermore, in order to ensure safety during stage movements when the objective is not retracted, the entire in-vacuum assembly is grounded to the FIB/SEM chamber, therefore allowing for sensing stage collisions. To focus, we use a piezoelectric linear positioner with active feedback and a precision better than 10 nm, which provides fine focus control and high repetivity. Considering the geometry of the vacuum chamber, in order to convey the beam-path normal to the glass window at the vacuum flange, we had to steer the beam at an angle of 17.2 degrees which imposed the use of a 3 × 1' mirror. Here, in order to prevent mechanical distortions of the mirror, the holder was designed to exert an equal pressure across the rim of the mirror. The design allows for use of high NA objectives such as the Olympus MPLAPON 50x/0.95. We tested multiple objectives for vacuum compatibility and found that degassing will occur for an average of 24 hr from the first pumping cycle (see below for details). Although we have not noticed any change in the optical performance of the objectives as a consequence of their permanence under high-vacuum, we expect that the cement used in the assembly of those complex optical elements will degrade over time and the objective will probably need to be replaced (our current experience describe this as >12 months).

Outside the vacuum chamber, the LM is composed of an aluminium body that is directly attached to the flange and guides the position of the required optical elements. In front of the glass window on the flange, a rectangular mirror steers the beam and ensures a $n(\frac{\pi}{2})$ tilt (where $n \in Z^+$) relative to the orientation of the objective optical axis. The microscope body has two arms, the one devoted to the illumination contains the fibre-end and the illumination tube-lens; while the detection arm hosts the imaging tube-lens and the camera. The body includes a dichroic selector, which in our case allows for four channel fluorescence (DAPI, GFP, RFP, Far-red) and reflected light illumination. The possibility of having a reflected bright-field illumination image can simplify correlation in cases where there are no fiducial beads or where the fluorescence signal cannot be directly matched to the shape of the sample. The current design allows for wide-field imaging, but Structured-Light-Illumination (SIM) (*Gustafsson, 2000*) could be implemented by modifying the excitation arm to accommodate the polarising optics and either a transmission grating or a programmable reflective ferroelectric liquid-crystal-on-silicon (LCOS) micro-display (*Kner et al., 2009*; *Shao et al., 2011*). The implementation of SIM would add optical sectioning and super-resolution capabilities.

In order to control all imaging modalities from the same computer and to have an interface designed around the needs of CLEM (*Figure 3*), we developed a program in Labview and Python. The interface allows for (i) simple image acquisition in all modalities; (ii) seamless navigation between modalities which can bring the same ROI centred relative to each beam and (iii) provides hierarchical data storage to simplify post-processing. This control interface, which we called PIE-scope-commander is freely available from the laboratory resource webpage (https://www.demarco-lab.com/resources).

## Setting up and using the PIE-scope

The first step, after the LM has been mounted and aligned, is to measure the exact position of the optical axis relative to the coincidence point of the FIB/SEM. This step requires a fiducial that can be recognised in all modalities. We noticed that, due to mechanical tolerances, every time the system is mounted there is a variability of up to 500 μm in the final position of the light microscope. The calibration is typically performed manually by estimating the shift between the coincident point and the expected position and adjusted by centring the fiducial. The shift values are then inserted in the control software in order to automatically perform direct movements between imaging modalities.

In the software, we assumed three positions (see *Figure 1* and *Video 1*), which can be customised: (i) LM imaging, where the sample orientation is normal to the optical axis of the LM; (ii) FIB imaging, where the sample orientation is normal to the optical axis of the FIB; (iii) lamella preparation, where the sample is at 15–22 degrees from the optical axis of the FIB. Once the imaging positions have been defined it is possible to perform pre-computed relative movements (shifts, tilt, and compucentric rotations) in order to image the same ROI under different modalities. The control software interface allows direct control of the FIB/SEM, a visual comparison between the FIB/SEM and LM images and saving the data in organised sub-directories.

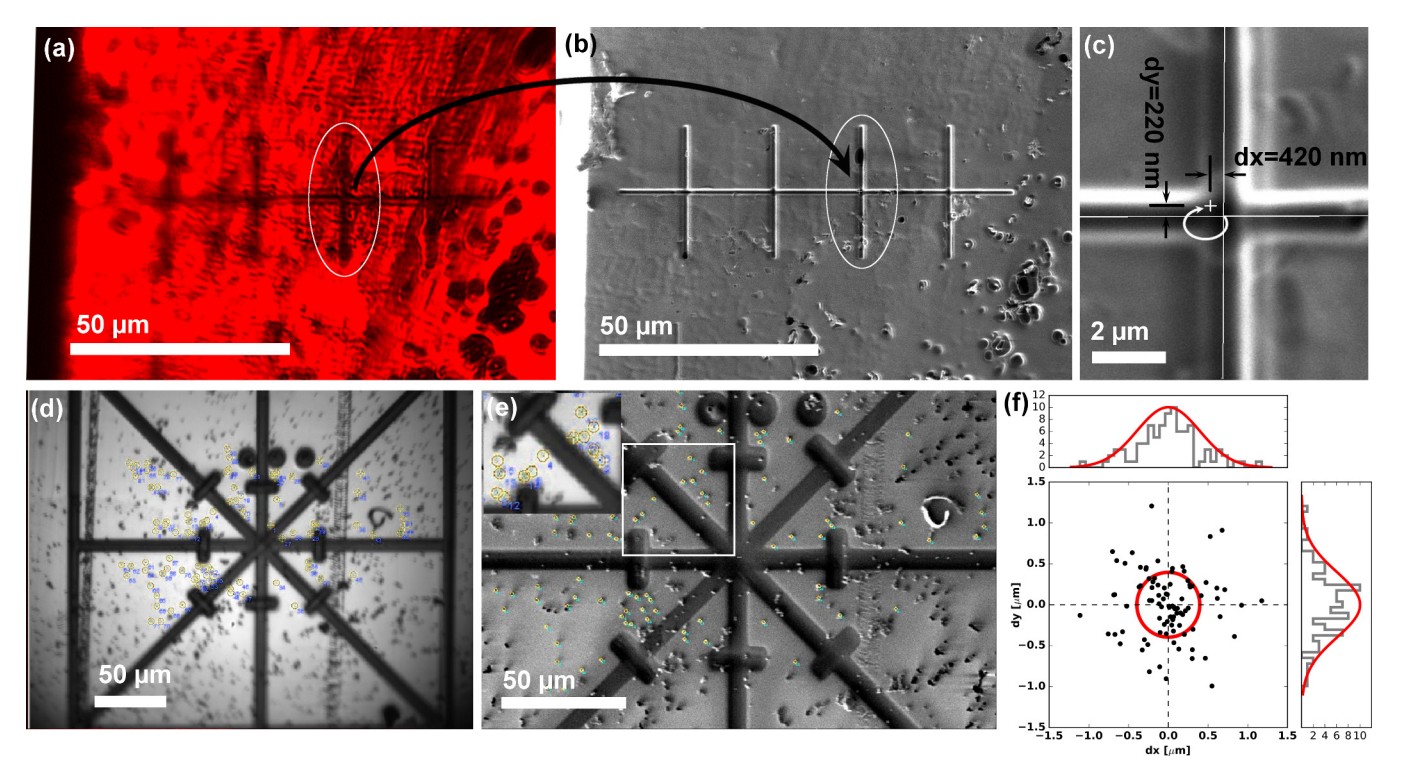

**Figure 3.** Translation and correlation precision of the integrated PIE-scope. (**a,b**) Optical (**a**) and FIB (**b**) image of the same sample region, containing a series of pre-milled test patterns (crosses). After the stage translation calibration and targeting steps one can select the target feature using the optical microscope (**a**), and simply translate the same field of view under the FIB and the (**b**). (**c**) The precision of direct, unsupervised ROI targeting by simple sample translation is <420 nm for the X-axis (major axis of translation) and <300 nm for the orthogonal Y-axis. (**d, e, f**) The procedure used to quantify the mechanical correlation precision when moving between an optical (**a**) and FIB (**b**) images. Fiducial markers were identified in all modalities and the transformation to perform the correlation was calculated. In (**f**) a scatter plot describing the mean residual for all the measurements (n = 86), showing the precision is ~500 nm (±σ). For every measurement, we selected beads > 100 beads to eliminate/average out eventual fiducial selection errors.
DOI: https://doi.org/10.7554/eLife.45919.004

In PIE-scope we implemented a basic correlation procedure, which leads the user to identify the location of the region of interest in the FIB or SEM image. Although using the proposed procedure is optional and specific use-cases might benefit from custom designed image processing, we find that the availability of a general method already present and embedded in the software greatly enhances the usability. The PIE-scope correlation is performed through custom made python scripts that allow selecting multiple points on the LM and FIB/SEM images to calculate the appropriate transformation. 2D correlation is performed simply by applying an affine transformation that includes anisotropic scaling to match the pixel spacing resulting from imaging a sample from different tilts. This procedure is best suited for 2D correlation and, according to previous reports (*Kukulski et al., 2011*; *Schorb and Briggs, 2014*), it can lead to correlation precisions which are better than 100 nm. If 3D correlation is required, this can be performed using the algorithm previously described in *Arnold et al. (2016)*, which can be downloaded

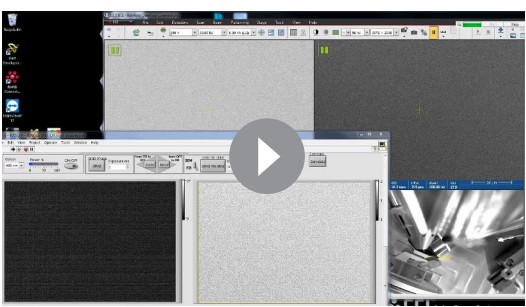

**Video 1.** PIE-scope operation. This video provides an overview of the capabilities provided by PIE-scope and provides an overview of its capabilities.
DOI: https://doi.org/10.7554/eLife.45919.005

as Python package from 3DCT (https://github.com/coraxx/3DCT). 3DCT can be used independently or it can be called from PIE-scope by modifying the call-back function of the correlate button in the LabVIEW GUI to interface with the 3DCT package.

## Characterising the system

Since we modified the vacuum system of the FIB/SEM we first ensured that the vacuum system performance was unaffected. This was measured using both the vacuum gauge of the FIB/SEM (HVG), and through the use of a residual gas analyser (from Pfeiffer). Once satisfied that both modified flanges did not present any leak we took advantage of the residual gas analyser to identify the average degassing time for the objective. This redefined the minimum pumping cycle length prior to insertion of cryo-samples to >20 hr. After this time, the vacuum in the FIB/SEM chamber is 1e-4 Pa at room temperature and 5e-5 Pa when the system is at cryogenic temperature. The condensation rate on cryo-samples was found to be identical to the rate measured before the installation of the PIE-scope. It is worth to mention that the maximum vacuum level obtainable varies from instrument to instrument and that the pumping time is dependent on the components present in the vacuum chamber. Therefore, the evaluation of the vacuum performance (or loss of performance) should be done on each individual instrument (e.g. Thermo Fisher Helios systems in average have better vacuum than Thermo Fisher Scios), and there should be no variation on the maximum vacuum level that can be reached after the PIE-scope installation.

The addition of an external component to the FIB/SEM leads to the potential introduction of vibrations into the system, which may result in mechanical drift. We measured the drift of the sample at room and cryo temperature with and without the light microscope installed to ensure no significant variation would be detected. With the presented hardware composition (fibre coupled Toptica iChrome-CLE and heated Basler USB3 camera) the drift remained within the specification of the FIB/SEM (<10 nm/min at room temperature and <30 nm/min at cryogenic temperature). Further, when imaging at the maximum framerate achievable with the SEM (50 frames/s) we have not detected changes in the vibrations.

Since the system is designed for correlative studies, the first feature to characterise was the repetitivity of the relocation precision between the LM and the coincidence point. As displayed in *Figure 4*, performing direct unsupervised movement shows that the relocation precision is <420 nm along X and <220 nm along Y. The relocation precision is a characteristic of the FIB/SEM stage and could vary from instrument to instrument, but most manufacturer's specifications claim a relocation precision better than 2 µm. Considering that the simplest use-case consists of positioning a 5–10 µm lamella around a target, the above-mentioned stage accuracy is sufficient. The relocation precision was measured by repeatedly moving between the FIB and the LM imaging positions (*Figure 4*). After each movement, we imaged the sample and calculated the residual error in relocation by calculating the required transformation to match the last image with the previous one (the residuals for each measurement are displayed in *Figure 4*). As previously shown (*Arnold et al., 2016*; *Schorb and Briggs, 2014*), when using fiducial markers, it is possible to improve the correlation accuracy to <100 nm despite the diffraction limit of the light microscope equipped with a non-immersion objective will be limited to ~320 nm in XY and ~555 nm in Z (assuming a NA of 0.95 and a wavelength of 500 nm). In order to quantify the optical performance of the system, we measured the maximum imaging resolution through the calculation of point-spread-functions. As expected, given the simplicity of the optical design, for all objectives tested we could measure a resolution in line with the theoretical diffraction limit. Through the use of image deconvolution, the LM resolution in all three dimensions could be mildly improved but in order to better resolve the molecular distribution at subcellular level super-resolution imaging would be better suited. The field of cryo-super-resolution fluorescence microscopy is still at its early stages and the most suitable method still needs to be identified but recently multiple approaches have been demonstrated (*Moser et al., 2019*; *Tuijtel et al., 2019*; *Xu et al., 2018*). The PIE-scope design is per se compatible with any type of epi-fluorescence imaging. Accordingly, if super-resolution imaging is required the implementation will be possible and will only require changes on the atmospheric side of the device. Along these lines, the effectiveness and usability of cryo-super-resolution imaging are predominantly limited by the ice contamination that inevitably builds up during long imaging sessions and by drift present in those systems (*Briegel et al., 2010*; *Wolff et al., 2016*; *Xu et al., 2018*). The PIE-scope resolves both these problems since the sample is kept in high vacuum during the entire imaging session and

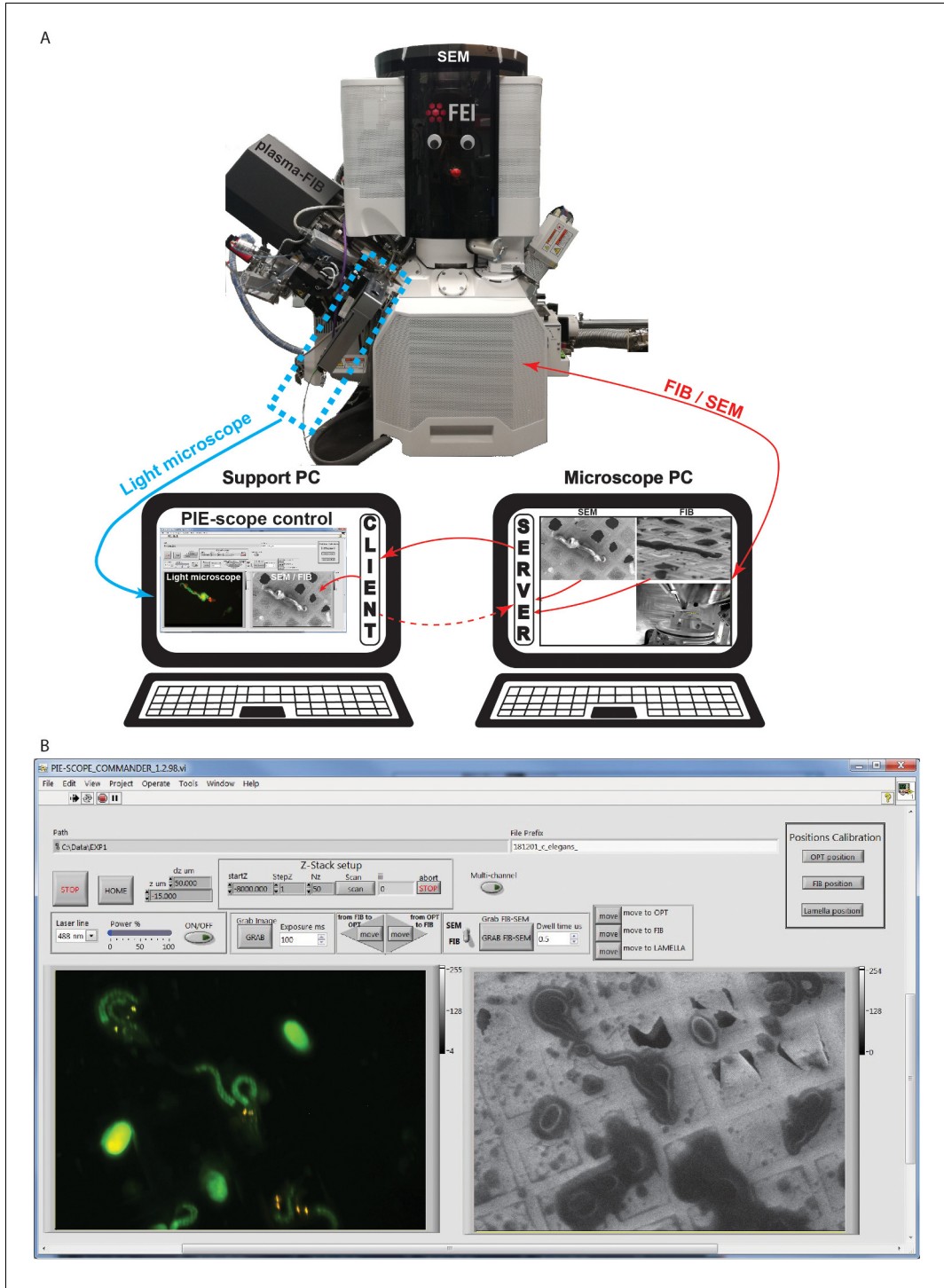

**Figure 4.** PIE-Scope commander, a CLEM-friendly interface. (**a**) A scheme of the communication setup. Here the microscope computer controls the FIB/SEM and runs the python control server. On the support computer, we run a python client (ThermoFisher Autoscript) and the custom built PIE-scope commander to control both the FIB/SEM and the LM. (**b**) The PIE-scope commander interface, all the required imaging positions can be calibrated from the main window, basic imaging parameters can be set, both FIB and LM images are visible side-by-side and data location can be defined for each experiment. File naming will have a user-defined root and appendix describing the imaging parameters (e.g. FIB, SEM, LM, laser line, dwell-time, exposure time).

DOI: https://doi.org/10.7554/eLife.45919.006

the drift rate is determined by the specifications of the cryo-stage in use, which must be compatible with high-resolution SEM imaging and FIB micromachining.

Finally, one aspect to consider if one wishes to use commercial objectives with an NA >0.8 is that the orientation and the position of the sample when mounted on the cryo-stage must allow imaging with sub-mm working distances. Luckily, all available cryo-stages come with inserts that can easily be modified or re-designed according to this specification if required.

## Cryo-CLEM on biological samples

In order to evaluate our new system on a real-life sample, we prepared targeted cryo-lamellae where the milling position was defined entirely through the fluorescence signal. Here, we used larvae of the nematode *Caenorhabditis elegans*, where four head neurons (BAGL/R and ASGL/R) express mCherry, to prepare a lamella centred around a neuronal cell body and one of the main dendrites which extend anteriorly towards the mouth of the animal. In *Figure 5* we show stills through the milling procedure, from the identification of the ROI, through the controls while thinning, to the final image after polishing. As shown in *Figure 5*, the fluorescence is maintained after milling, which can be used to confirm that the target structure (a dendrite of a BAGL/R neuron) is effectively contained in the lamella.

We further tested the ability of targeted lamella preparation in a bulk cryo-sample. Here a whole brain dissected from a *Drosophila melanogaster* larva expressing EGFP in interneurons (Engrailed-Gal4, UAS-eGFP) was cryo-fixed and imaged under fluorescence. Based on the fluorescence signal, 1.5 μm thick lamellas were isolated around the neuronal body (*Figure 6*). This setup is compatible with emerging techniques such as the cryo-lift-out which will allow selective isolation of lamellae from bulk tissue (*Mahamid et al., 2015*; *Parmenter et al., 2016*; *Rubino et al., 2012*; *Zhang et al.,*

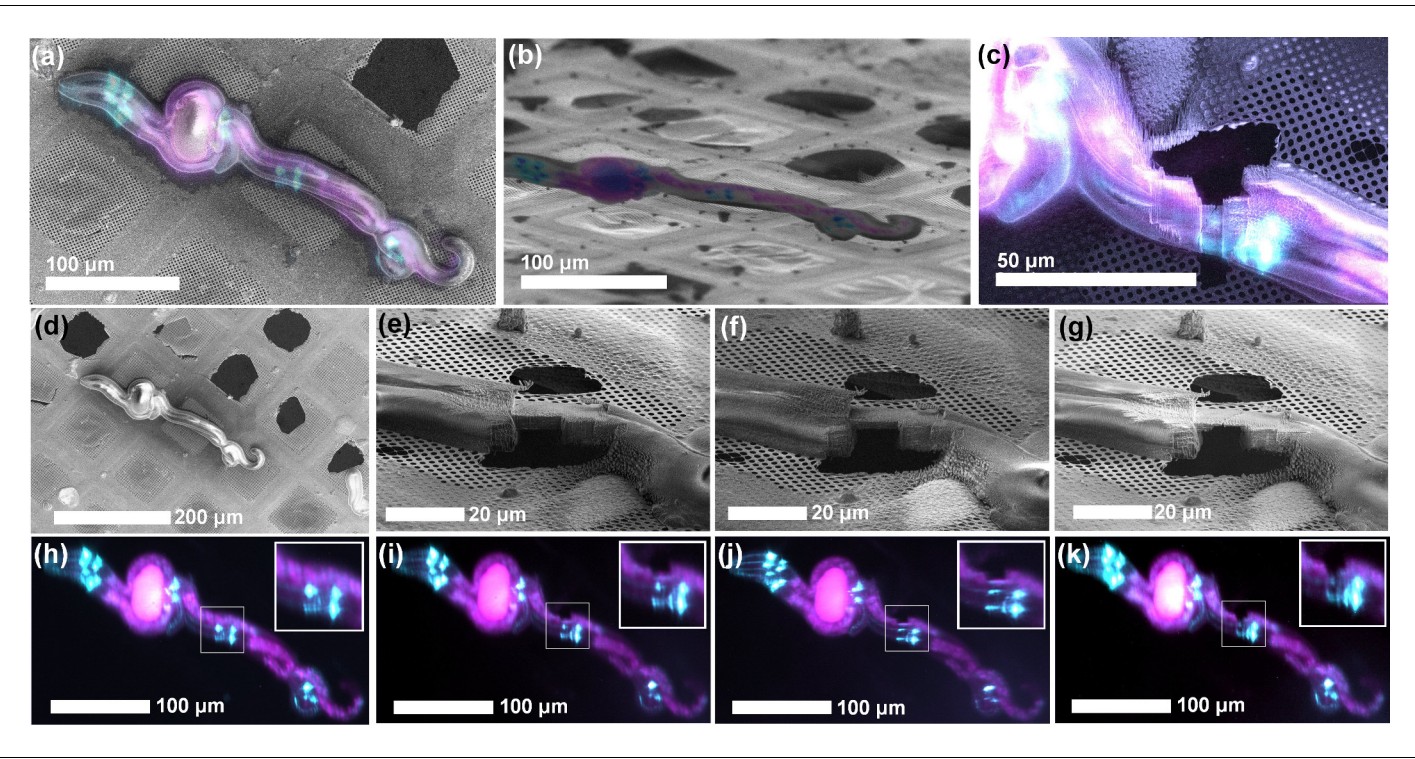

**Figure 5.** ROI identification, targeting and cryo-lamella preparation in *C. elegans* larvae. (a) Correlation between the SEM (top view) and fluorescence microscopy images. (b) Correlation between the FIB and fluorescence microscopy images. The sample is tilted into the lamella preparation position (22° with respect to the grid plane). (c) SEM image of the thinned down lamella overlayed with the fluorescence image from the same region. (d,h) SEM and optical images, respectively, of the initial state of the sample, while the pairs (e, i), (f, j) and (g, k) show SEM and optical images, respectively, of the intermediate steps in the lamella, thinning. Since the optical microscope is integrated within the system, multiple ROI-verifications are possible throughout the workflow without the need for cryo-transfers.

DOI: https://doi.org/10.7554/eLife.45919.007

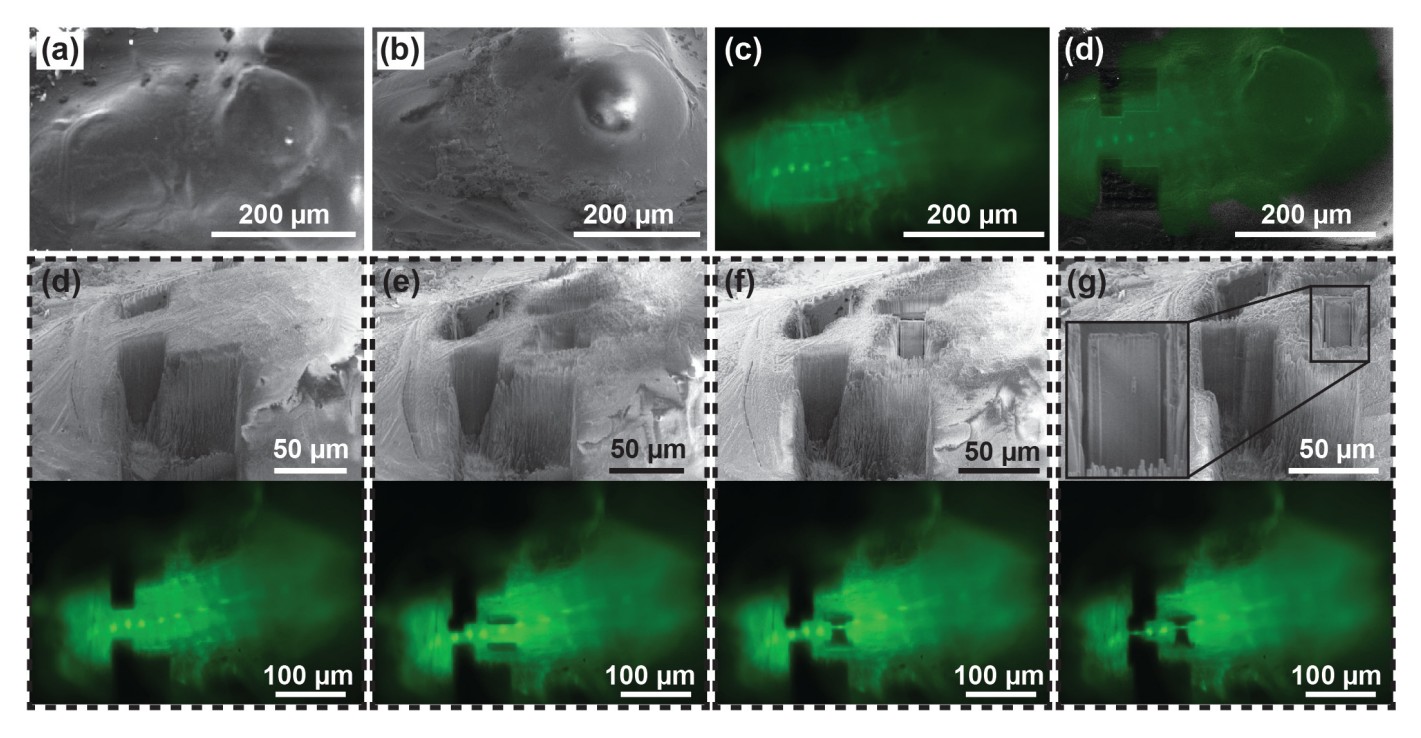

**Figure 6.** Targeted lamella preparation in a bulk cryo-sample (larval *D. melanogaster* brain expressing EGFP in interneurons). (a, b, c) Initial FIB (a), SEM (b) and LM (c) images. (d) Milling of straight edges in the sample allows improved precision of correlation between images acquired in different modalities. Here, the overlay between LM and FIB images of the pre-milled sample is shown. (d, e, f, g) Pairs of SEM and optical microscopy images of the initial sample state (d), two intermediate verification steps (e and f), and the final result (g). Based on the fluorescence signal, 1.5 µm thick lamellae were isolated around the neuronal body (g). Inset in (g) shows a magnified image of a lamella ready for a cryo-lift-out.
DOI: https://doi.org/10.7554/eLife.45919.008

*2016*). The position of the LM does not interfere with the manipulator, making it compatible with the development of future applications involving cryo-lift-out.

Considering the sample size, in both the *C. elegans* larvae and the *D. melanogaster* brain the lamellas were kept of thicknesses above 1 µm. This thickness still allows high-quality fluorescence imaging because the intensity in the lamella and in the uncut tissue have comparable values. When a lamella has a thickness compatible with cryo-TEM, typically the intensity is 30% of the one measured in a focal plane within the uncut tissue. This value is in line with the difference between the average thickness of a lamella (~250 nm) and the depth of focus of the objective we use (~800 nm). Therefore, in the absence of an optical sectioning method such as confocal microscopy, the limited dynamic range of the detectors make imaging of thin lamellas challenging. Nevertheless, in order to show that this is possible, we prepared cryo-TEM compatible lamellas on Yeast a strain stained with a nuclear stain (Nuclear-ID, EnzoLife Sciences). In *Figure 7* it is possible to see that fluorescence is detectable in a lamella, and further that the fluorescence signal can be used to target the cryo-ET data collection. In this case, the fluorescence was diffused through the nucleus and we used Cryo-ET analyses (*Figure 7e* and *Video 2*) show that LM imaging does not induce de-vitrification of the sample.

## Conclusions

We have presented a novel system to perform cryo-CLEM and targeted cryo-lamella preparation, which simplifies sample handling and reduces the risk of sample contamination by limiting the number of cryo-transfers. Furthermore, having a light microscope within the vacuum chamber provides an opportunity to verify that the target has not been lost while milling. Future developments will likely involve the introduction of optical sectioning and super-resolution imaging. Further, having the light microscope on the same stage as the FIB resolves the current status where each microscope

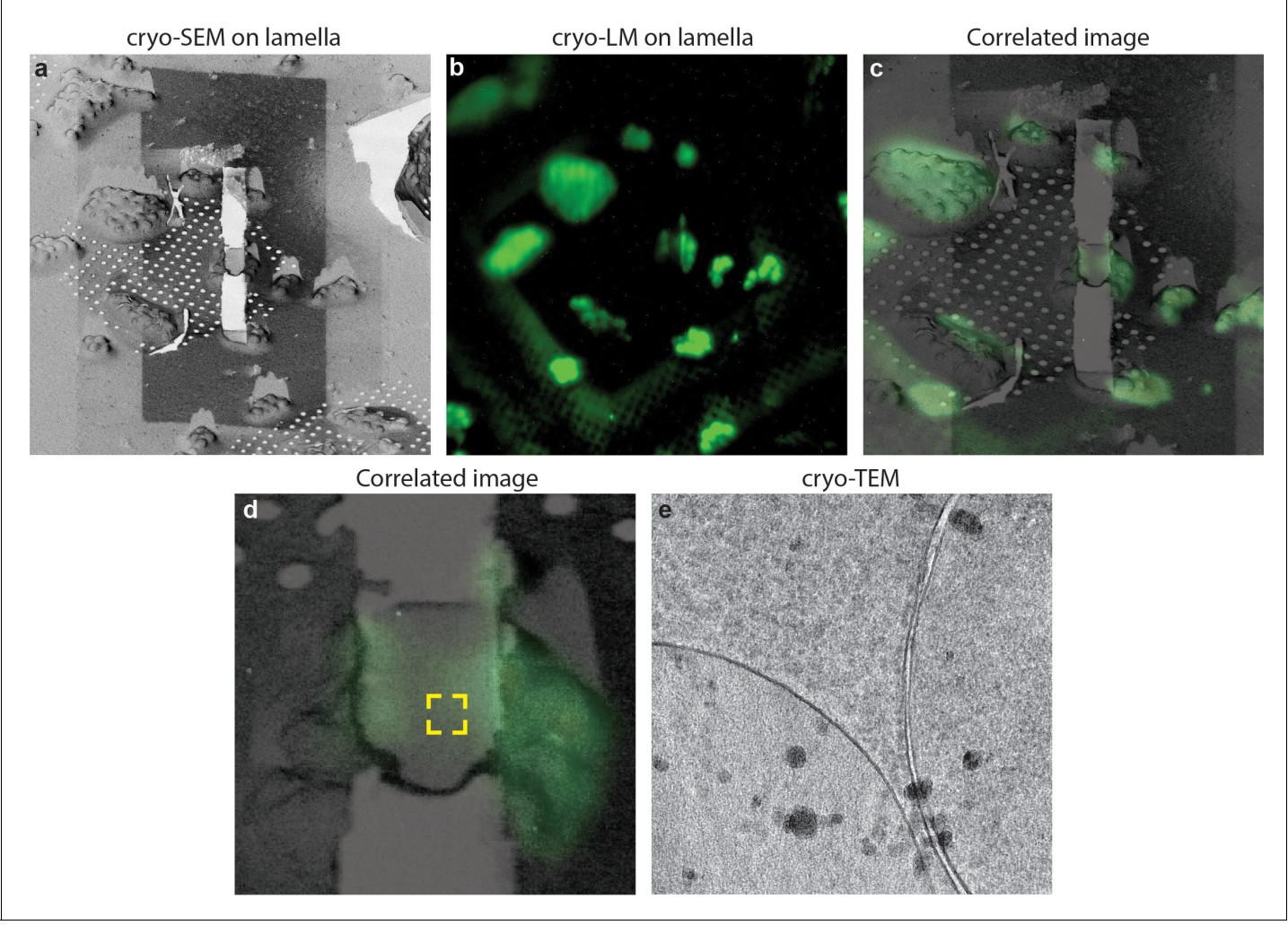

**Figure 7.** Cryo-correlative microscopy on fluorescent yeast cells. (**a–b**) *S. cerevisiae* cells were labelled with nucleolar staining, lamellas were prepared and then imaged using cryo-fluorescence microscopy. (**c**) Correlated image between the SEM and LM images, the region imaged through cryo-ET is labelled in the yellow square. (**d**) Cryo-TEM image of the lamella showing that the sample is still vitreous.

DOI: https://doi.org/10.7554/eLife.45919.009

comes with its own image format and no stage information can be directly imported from the LM to the FIB/SEM. This configuration provides the foundation required for automated CLEM to mill targeted cryo-lamellae, which will enable structural studies on rare cellular events.

Considering future software requirements, it is reasonable to expect setups which include automated segmentation of correlated images to define or guide the definition of the milling parameters with limited user intervention. When using the conventional suite of separated microscopes it is challenging to achieve such a target because each microscope has its own image format and no stage information can be directly imported from the LM to the FIB/SEM. In the PIE-scope, all images are acquired using the same stage and software (*Figure 3b*), and fluorescence images can be acquired in between milling steps (therefore providing additional information useful for a more precise correlation, such as the milled edges). Through the PIE-scope, it will be therefore possible to implement CLEM automation for targeted lamella preparation.

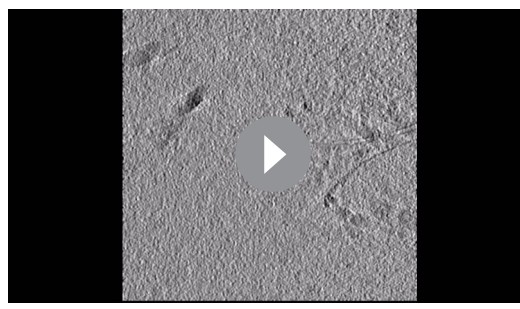

**Video 2.** Cryo-electron tomogram on a lamella after fluorescence imaging. The region imaged on this tomogram correspond to the one visible in *Figure 7d*. no de-vitrification is visible as a result of LM imaging.
DOI: https://doi.org/10.7554/eLife.45919.010

## Materials and methods

### The PIE-Scope setup

The illumination source is a four wavelength combiner with a single multimode fibre output (Toptica iCLE-50 405, 488, 561, 640 nm). The choice between a multi-band-pass dichroic (Chroma 89402 m) and a Thorlabs 10/90 (R/T) beam splitter allows for fluorescence or reflected light imaging. The maximum laser power used during imaging corresponded to 0.5 mW out of the fibre, which was further reduced to 0.2 mW at the back-focal plane of the objective, although there might be occasions where more laser power is required for imaging, we noted that these conditions have provided sufficient illumination on all tested samples. Currently, the camera used is a Basler acA1920-155um with USB3.0 interface. This camera is a heated CMOS detector with a quantum efficiency in the visible range of ~70%, it represents a cost-effective solution which comes with a dark noise of 6.6 e⁻. The major advantage of this component, beside its price, is that it does not require active cooling, making the system less susceptible to vibrations. If single-molecule imaging is required, or if the fluorescence signal is extremely dim, then cooled CMOS cameras such as the Hamamatsu Flash4.0 V3 should be used. In this case, in order to ensure that no vibration is introduced in the system, the camera cooling lines should be connected to the FIB/SEM water chiller. The feedthrough from the atmosphere to the high-vacuum is realised through a glass window (Thorlabs VPCH42-A). The optical path is steered using three axes mirror holders embedded in the custom build microscope body/flange visible in *Figure 2a*. The focus drive is a high-vacuum-compatible piezoelectric linear positioner (Smaract SLC-2445HV). The objectives tested were Olympus MPLFNL 10x/0.3, MPLFNL 20x/0.45, UPFLN 20x/0.8 and MPLAPON 50x/0.95.

The microscope body and the adapters were machined at the Monash University Mechanical and Aerospace Engineering Workshop. All atmosphere parts were made of Aluminium alloy 7075. All vacuum parts were machined from Stainless Steel 316 l. Mounting of the microscope body on the vacuum chamber of the FIB/SEM occurs directly on the middle front GIS port (GIS2 on ThermoFisher instruments). The only strict requirement of this design is that neither a gas injection system (GIS) nor detector can be mounted on the front GIS ports at the FIB side of the chamber, we resolved this issue by repositioning our GISs from the flanges GIS1 and GIS2 to the rear flanges (GIS4 and GIS5). Although this can be a limitation for highly equipped material science microscopes, on systems where biological imaging and lamella cryo-preparation is the major focus this will not result in a problem. Also, with the advent of multi-line-GISs offered by most instrument manufacturers, it is extremely unlikely that all GIS flanges will be required.

All custom components were designed using Autodesk Inventor Pro 2016 and Siemens NX. Parts were fabricated in the Mechanical and Aerospace engineering mechanical workshop at Monash University. Ray tracing was performed using Zemax OpticsStudio. The control software has been developed using National Instruments LabVIEW 2015, Python three with Numpy 1.15, Scipy 1.1.0 and ThermoFisher AutoScript 4.0. All drawings, latest Bill of Material (matched to the stable software released), a stable software release, list of dependencies and instructions are available at https://www.demarco-lab.com/resources.

### Sample preparation

All *Caenorhabditis elegans* strain RJP3088 (Pets-5::mCherry +Pelt-2::GFP) (*Juozaityte et al., 2017*) was maintained at 20°C on NGM plates seeded with *Escherichia coli* OP50 bacteria (Porta-de-la-Riva, Fontrodona et al.). Isolation of L1 larvae was performed as previously described (*Porta-de-la-Riva et al., 2012*). Larvae were deposited on a holey carbon EM-grid and plunge-frozen in liquid ethane. *C. elegans* samples were kept in M9 medium, concentrated to a density of 3–4 worms / μL.

5 µL of solution were deposited on a Cu-300-mesh Quantifoil R-2/2 grid which was glow discharged for 30 s using a Pelco EasyGlow. The excess of medium was manually blotted using Whatman filter paper grade 595 (Merck) and vitrified by plunge freezing in liquid ethane.

*Drosophila melanogaster* used were a cross between Engrailed-Gal4 (BL30564) and UAS-eGFP (BL6874) from the Bloomington Drosophila Stock Center (Indiana). Flies were raised on standard media at 25°C. Drosophila brains at the third instar larval stage were dissected in cold phosphate buffered saline before being mounted in 20% (w/v) dextran solution prior vitrification. Samples were high-pressure frozen using a Leica EM-PACT2 on 200 µm deep planchettes. In order to expose the sample to the surface, therefore reducing the milling time required, we performed manual freeze-fracture by scraping the surface with a scalpel under liquid nitrogen. The difference in density between the dextran solution and the sample itself ensures an easy separation.

*Saccharomyces cerevisiae* cells were cultured at overnight 30°C in 2 YPD+. Cell growth was monitored through the measurement of the optical density. Cells were harvested when the optical density reached 0.6, washed in PBS and stained using Nuclear-ID (Enzo Life Sciences). 5 µL were deposited on a Cu-300-mesh Quantifoil R-2/2 grid which was previously glow discharged for 30 s in a Pelco EasyGlow. Grids were manually blotted for 5 s using Whatman 595 blotting pads and plunge-frozen in liquid ethane.

## Acknowledgements

We acknowledge Tom Nichols and Kenny Mani (Thermo Fisher Scientific) for fruitful discussions on the setup. Hari Venugopal (Monash cryoEM platform) for the support on the Titan Krios.

## Additional information

### Funding

| Funder | Grant reference number | Author |
| --- | --- | --- |
| ARC Centre of Excellence in Advanced Molecular Imaging | CE140100011 | Sergey Gorelick Genevieve Buckley James C Whisstock Alex de Marco |
| Australian Research Council | FL180100019 | James C Whisstock Travis K Johnson Alex de Marco |

The funders had no role in study design, data collection and interpretation, or the decision to submit the work for publication.

### Author contributions

Sergey Gorelick, Data curation, Software, Formal analysis, Validation, Investigation, Writing—review and editing, Collected and analysed the data, coded the control software, conceived and built the setup; Genevieve Buckley, Software, Visualization, Methodology, Writing—review and editing, Coded the control software; Gediminas Gervinskas, Investigation, Collected and analysed the data; Travis K Johnson, Monica Pia Caggiano, Investigation, Prepared the samples; Ava Handley, Investigation, Writing—review and editing, Prepared the samples; James C Whisstock, Funding acquisition, Writing—review and editing; Roger Pocock, Formal analysis, Investigation, Writing—review and editing; Alex de Marco, Conceptualization, Supervision, Funding acquisition, Writing—original draft, Project administration, Collected and analysed the data, conceived and built the setup

### Author ORCIDs

Roger Pocock (iD) http://orcid.org/0000-0002-5515-3608
Alex de Marco (iD) https://orcid.org/0000-0001-6238-5653

### Decision letter and Author response

Decision letter https://doi.org/10.7554/eLife.45919.015
Author response https://doi.org/10.7554/eLife.45919.016

## Additional files

### Supplementary files
• Transparent reporting form
DOI: https://doi.org/10.7554/eLife.45919.011

### Data availability

All significant data are included in this manuscript. The mechanical drawings for all of the components, the bill of material, and the control software have been made available through the laboratory webpage (https://www.demarco-lab.com/resources) and through Zenodo (10.5281/zenodo.3260173).

The following dataset was generated:

| Author(s) | Year | Dataset title | Dataset URL | Database and Identifier |
|---|---|---|---|---|
| Sergey Gorelick, Genevieve Buckley, Gediminas Gervinskas, Travis K Johnson, Ava Handley, Monica Pia Caggiano, James C Whisstock, Roger Pocock, Alex de Marco | 2019 | PIE-scope, integrated cryo-correlative light and FIB/SEM microscopy | https://dx.doi.org/10.5281/zenodo.3260173 | Zenodo, 10.5281/zenodo.3260173 |

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
