## [Decision Letter]

Thank you for submitting your article "PIE-scope: integrated cryo-correlative light and FIB/SEM microscopy" for consideration by *eLife*. Your article has been reviewed John Kuriyan as the Senior Editor, Sriram Subramaniam as Reviewing Editor, and three reviewers. The following individual involved in review of your submission have agreed to reveal their identity: Jürgen M Plitzko (Reviewer #1).

The reviewers have discussed the reviews with one another. You will see that all three reviewers find your work interesting but raise a number of concerns. Since each reviewer's concerns have a somewhat different flavor, we have chosen to send the individual reviews to you below without merging them. Your revised submission must satisfactorily address the major comments and concerns of the reviewers in order for the manuscript to be considered for publication in *eLife*.

*Reviewer #1:*

The manuscript "PIE-scope: integrated cryo-correlative light and FIB/SEM microscopy" corresponded by Alex de Marco describes a "triple-beam" instrument that includes photons, ions and electrons, referred to as a "PIE-scope" for the correlation and fabrication of cryo-FIB lamellae from frozen hydrated and high-pressure frozen samples for subsequent analysis by cryo-electron tomography (cryo-ET). The integration eliminates the multiple sample transfers that typically take place in the current cryo-ET workflow, which greatly simplifies its application. As a result, and through its retrofittable design, it can provide a way to make this workflow more accessible to other laboratories while increasing the overall throughput. The authors plausibly document their developments and achievements, and I regard this as important advance in this field. However, while the manuscript is direct and to the point, its writing appears somewhat rushed and misses details. In some places the writing is too technical (or jargon) and therefore it could be definitely improved to make it more accessible.

It is somewhat disappointing that there is no final TEM analysis showing that after correlation and sample preparation, cryo-ET can be successfully performed on the produced lamella from either of the two samples. The question is: Could this still be performed?

The paper by Gorelick et al., presents a new design for an integrated fluorescent and dual-beam microscope. This technological advance is a great addition to the correlative tools available for cellular cryo-electron tomography. The implementation described is simple and elegant and seems to work very well. In particular, the ability to target with a precision of ~500 nm is very encouraging.

I find the technology worthy of publishing, but I think that the manuscript requires significant revisions in order to target a wide readership, but more importantly more results need to be included to make sure it will be a useful tool for people interested in this technology. Besides needing many more details in the text, a manuscript like this should contain detailed supplementary data for the expert reader.

Reviewer #2:

1) Information necessary to reproduce the setup.a) The authors state that the software is constantly changing and can be made available from the authors. In order for anyone to implement this technology, they have to be sure that the work described here can be implemented outside of the authors' set up. Thus, a stable release of the software, with a detailed description of all the requirements, should be a necessary condition for publication.b) Along these lines, the drawings and all needed equipment for reproducing the set up should be available (can be upon request but needs to be explicitly stated).

2) The imaging is only done in 2D. The authors describe three positions in software: LM imaging, FIB imaging, and lamella preparation.a) How are the transformation of coordinates done for 2-D images between these imaging modes? Specifically, how is the fluorescence signal superimposed onto the tilted (FIB milling) position? Is the rotation simply done by a compression in the direction of the tilt?b) What is the SEM imaging position with respect to the others? A table, figure, or paragraph describing the relative distances and angles should be provided.c) When checking for the fluorescent signal at different stages of milling, is the sample tilted back so that it is orthogonal to the FIB beam, or is the tilted image used? Do the authors routinely use FIB imaging during milling, or is only the SEM used for this task?

3) In describing the sample preparation, the authors are not very explicit. A notable example is the lack of details in the brain sample. Was it high-pressure frozen? If it was plunge frozen, there is no chance that this sample will be vitrified. While the authors may argue that for the purposes of this manuscript this is not relevant, it should then be clearly stated.

4) The biggest gap in the manuscript is the characterization of the quality of the lamellae for cryo-ET. Each of the points are essential for the work to be useful:a) While the authors did not alter the cryo-stage, the authors should test and report on whether there are any alterations to the stability of the system in any way. Are there any new vibration or other issues that were introduced?b) Are there any added contamination rates from the introduction of the fluorescence set up? Without subsequent cryo-TEM images, it is hard to gauge the quality of the finished lamella.c) In fact, the finished lamella seems thick and cracked, and the thickness is not reported. Even if one would consider this work not to be about lamellae production, the detectability of the fluorescent signal for very thin lamellae should be demonstrated.d) A major issue with fluorescence imaging at cryogenic temperatures is the potential for devitrification. If the authors do not think this is a concern due to the low power of the illumination used, they should provide clear arguments. otherwise, they should show the vitrification of the lamella in the TEM after preparation in the PIE scope.

5) In their letter, the authors claim that this is a non-expensive set up ($50K). I think that as part of supplementary data they should provide a list of materials required and their approximate cost, which is customary when describing a new set up. For instance, what is the cost of ThermoFishder Autoscirpt?

*Reviewer #3:*

General assessment:

Very well written. Clearly described what has been reached and how the system provides a significant addition to the set of tools available for cell biological studies that require morphological/structural information of molecular structures in their cellular context.

The capability to perform the preparation of cryo-lamella using a focused ion beam SEM with an integrated light microscopy is very exciting as it can be used to ensure that the feature of interest is present in the lamella. This approach will improve possibilities to perform cell biological experiments in which targeted cryo electron tomography of fluorescently tagged structures are an essential ingredient.

It is very exciting that a proof-of-principle is provided on larvae of *C. elegans* on which the technological solution is evaluated. It is also exciting that the instrumentation is principle retrofittable and the software to make all this work for CLEM operation is open source.

Major concerns:

A major concern I have with the manuscript is that though it describes overall features and possibilities of the instrumentation with sufficient detail to illustrate the potential of its capabilities with the proof-of-principle, but that it lacks a discussion on some of the remaining technological bottlenecks of 3D CLEM on frozen hydrated specimen. In my view the steps that still need to be taken to meet the requirements to make the system applicable for a more wider audience are not mentioned with sufficient detail.

Subsection “Characterizing the system” summarizes the specifications of the system very nicely. Bottlenecks are the accurate 2D positioning of the lamella around a target (~200-400 nm, though is claimed that it could be ~~100 nm), the reduction in imaging quality due to drift (optical astigmatism due to drift, how large is the drift and can limitation be solved?) as well as the outlook for 3 D correlation to achieve an accuracy that would be sufficient for targeted FIB-milling (~500 nm, the need for deconvolution to improve the z-resolution can this indeed be improved, and to what accuracy?)

It would strengthen the paper considerably, if additional discussion/argumentation was giving detailing the possibility of a more accurate performance on next-generation systems.

---

## [Author Response]

Reviewer #1:The manuscript "PIE-scope: integrated cryo-correlative light and FIB/SEM microscopy" corresponded by Alex de Marco describes a "triple-beam" instrument that includes photons, ions and electrons, referred to as a "PIE-scope" for the correlation and fabrication of cryo-FIB lamellae from frozen hydrated and high-pressure frozen samples for subsequent analysis by cryo-electron tomography (cryo-ET). The integration eliminates the multiple sample transfers that typically take place in the current cryo-ET workflow, which greatly simplifies its application. As a result, and through its retrofittable design, it can provide a way to make this workflow more accessible to other laboratories while increasing the overall throughput. The authors plausibly document their developments and achievements, and I regard this as important advance in this field. However, while the manuscript is direct and to the point, its writing appears somewhat rushed and misses details. In some places the writing is too technical (or jargon) and therefore it could be definitely improved to make it more accessible.

We re-worded multiple parts of the manuscript in order to make it more accessible. Please refer to the text changes in the re-submitted document.

It is somewhat disappointing that there is no final TEM analysis showing that after correlation and sample preparation, cryo-ET can be successfully performed on the produced lamella from either of the two samples. The question is: Could this still be performed?

The analysis could not be performed on the samples presented in the first version of the manuscript because neither sample are available. As pointed out by the reviewer in the minor comments the size can be considered on the extreme side. We therefore decided to add a figure where we used yeast cells with live cell imaging compatible nuclear staining. Here we show that the sample is still vitreous after imaging with the light microscope and that successful targeted cryo-ET can be performed (Figure 7 and Video 2).

Reviewer #2:

The paper by Gorelick et al., presents a new design for an integrated fluorescent and dual-beam microscope. This technological advance is a great addition to the correlative tools available for cellular cryo-electron tomography. The implementation described is simple and elegant and seems to work very well. In particular, the ability to target with a precision of ~500 nm is very encouraging.I find the technology worthy of publishing, but I think that the manuscript requires significant revisions in order to target a wide readership, but more importantly more results need to be included to make sure it will be a useful tool for people interested in this technology. Besides needing many more details in the text, a manuscript like this should contain detailed supplementary data for the expert reader.1) Information necessary to reproduce the setup.a) The authors state that the software is constantly changing and can be made available from the authors. In order for anyone to implement this technology, they have to be sure that the work described here can be implemented outside of the authors' set up. Thus, a stable release of the software, with a detailed description of all the requirements, should be a necessary condition for publication.b) Along these lines, the drawings and all needed equipment for reproducing the set up should be available (can be upon request but needs to be explicitly stated).

We added supplementary files including a link to the lab resource webpage (https://www.demarco-lab.com/resources) with the stable release of the control software with the installation requirements, a bill of material comprising the required and optional (configuration specific) components and the drawings of the custom components. The drawing provided include only the custom components because the FIB-SEM chamber is proprietary intellectual property of ThermoFisher. Upon written request of the model to ThermoFisher and upon written confirmation from ThermoFisher that disclosure is acceptable the authors will be able to disclose the full assembly should this be required.

2) The imaging is only done in 2D. The authors describe three positions in software: LM imaging, FIB imaging, and lamella preparation.a) How are the transformation of coordinates done for 2-D images between these imaging modes? Specifically, how is the fluorescence signal superimposed onto the tilted (FIB milling) position? Is the rotation simply done by a compression in the direction of the tilt?

We agree that although the focus on the paper is on the hardware rather than the image correlation, a more detailed description of the correlation procedure is beneficial to help the readers. Accordingly, we added a paragraph to detail the correlation procedure, further in the Video 1 the correlation procedure is shown.

In short, we provide a tool to perform the 2D correlation which is embedded in the PIE-scope control software, which is based on calculation of the rigid-body transformation between the imaging modalities. If 3D correlation is required users can call external CLEM software such as 3DCT (We decided against including this package in PIE-scope is due to the impossibility for us to support it). The Video 1 shows the correlation procedure.

b) What is the SEM imaging position with respect to the others? A table, figure, or paragraph describing the relative distances and angles should be provided.

We modified Figure 1 to include the information about the relative position and orientation between imaging all modalities.

c) When checking for the fluorescent signal at different stages of milling, is the sample tilted back so that it is orthogonal to the FIB beam, or is the tilted image used? Do the authors routinely use FIB imaging during milling, or is only the SEM used for this task?

For fluorescence imaging the sample must be orthogonal to the FIB in order to access the sample with the short working distance of the LM objective. The FIB is used for imaging only the milling pattern must be placed. In both Figure 5 and Figure 6 all images were acquired using the SEM.

3) In describing the sample preparation, the authors are not very explicit. A notable example is the lack of details in the brain sample. Was it high-pressure frozen? If it was plunge frozen, there is no chance that this sample will be vitrified. While the authors may argue that for the purposes of this manuscript this is not relevant, it should then be clearly stated.

We extended the description of the sample preparation and included detail on the vitrification.

4) The biggest gap in the manuscript is the characterization of the quality of the lamellae for cryo-ET. Each of the points are essential for the work to be useful:a. While the authors did not alter the cryo-stage, the authors should test and report on whether there are any alterations to the stability of the system in any way. Are there any new vibration or other issues that were introduced?

The introduction of the light microscope in its current configuration has no effects on vibrations because the laser combiner is fibre coupled to the instrument and the lasers are solid state diodes which can be passively cooled. The camera is a heated detector, therefore no vibration due to cooling is introduced. The cryo-stage we have on the FIB/SEM has a drift speed limit of 60 nm / min. When first installed it demonstrated to drift 20-30 nm / min depending on the environmental conditions (room temperature fluctuations, multiple people around the instrument, etc.). Since the installation of PIE-scope we have never experienced a drift rate greater than 30 nm / min. Accordingly, a discussion has been added.

*b) Are there any added contamination rates from the introduction of the fluorescence set up? Without subsequent cryo-TEM images, it is hard to gauge the quality of the finished lamella.*

We have not measured any change in the contamination rate due to the installation of the light microscope. The reason is linked to the fact that the vacuum parameters do not change as a consequence of the installation. All feedthroughs used are high vacuum compatible and their performance is measured in respect to the best obtainable vacuum in the FIB/SEM before and after installation of PIE-scope. Since the LM objective is not designed for operation under high vacuum the degassing is slower than the other components. We added a paragraph to discuss this point, explaining that cryo-experiments should not be conducted if the vacuum chamber has not been pumped for at least 20 hours. The contamination rate is increased as consequence of slow degassing. Once the ideal vacuum conditions have been reached (which can be easily measured through the vacuum gauge on the FIB/SEM or through a residual gas analyser (if available) the experiments can start.

c) In fact, the finished lamella seems thick and cracked, and the thickness is not reported. Even if one would consider this work not to be about lamellae production, the detectability of the fluorescent signal for very thin lamellae should be demonstrated.

In order to address this point, we added a figure (Figure 7) which shows the fluorescence detection on a thin lamella. The thickness is demonstrated also through cryo-TEM. We also added a small discussion about the challenges presented when imaging a thinned lamella. We also added a short discussion relative to this.

d) A major issue with fluorescence imaging at cryogenic temperatures is the potential for devitrification. If the authors do not think this is a concern due to the low power of the illumination used, they should provide clear arguments. otherwise, they should show the vitrification of the lamella in the TEM after preparation in the PIE scope.

We agree with the reviewer that this is a major point to consider. We have provided additional data that include the cryoTEM after preparation using the PIE-scope. We further added the measurement of the total laser power used during imaging in the Materials and methods section.

5) In their letter, the authors claim that this is a non-expensive set up ($50K). I think that as part of supplementary data they should provide a list of materials required and their approximate cost, which is customary when describing a new set up. For instance, what is the cost of ThermoFishder Autoscirpt?

The cost of the hardware, including the lasers (which represent the vast majority if the cost) is below 50K USD (as described in the supplementary bill of material). It is worth mentioning that the cost can be reduced by ~20K USD if one choses to use a LED combiner or a mercury lamp with filters instead of the lasers (e.g. we identified the LED4D254 led combiner from Thorlabs as a good replacement, which including the driver module and light-guide will cost 5700 USD). Similarly, the cost can increase by 30K USD by integrating a more flexible laser combiner (such as the Coherent Galaxy) and a more sensitive camera such as the Hamamatsu Flash 4.0 V3. The integration using AutoScript is not mandatory, although it provides better control of the instrument, making the process smoother and the correlation faster. The average price of AutoScript is ~ 20K USD.

A license of LabVIEW is advisable (most universities have a site license) as it allows for further customization and development of the software, but it is not required for basic use. We provide both the source code and the compiled version of the stable release which only requires downloading of the LabVIEW runtime (which is free).

Reviewer #3:

General assessment:Very well written. Clearly described what has been reached and how the system provides a significant addition to the set of tools available for cell biological studies that require morphological/structural information of molecular structures in their cellular context.The capability to perform the preparation of cryo-lamella using a focused ion beam SEM with an integrated light microscopy is very exciting as it can be used to ensure that the feature of interest is present in the lamella. This approach will improve possibilities to perform cell biological experiments in which targeted cryo electron tomography of fluorescently tagged structures are an essential ingredient.It is very exciting that a proof-of-principle is provided on larvae of C. elegans on which the technological solution is evaluated. It is also exciting that the instrumentation is principle retrofittable and the software to make all this work for CLEM operation is open source.Major concerns:A major concern I have with the manuscript is that though it describes overall features and possibilities of the instrumentation with sufficient detail to illustrate the potential of its capabilities with the proof-of-principle, but that it lacks a discussion on some of the remaining technological bottlenecks of 3D CLEM on frozen hydrated specimen. In my view the steps that still need to be taken to meet the requirements to make the system applicable for a more wider audience are not mentioned with sufficient detail.Subsection “Characterizing the system” summarizes the specifications of the system very nicely. Bottlenecks are the accurate 2D positioning of the lamella around a target (~200-400 nm, though is claimed that it could be ~~100 nm), the reduction in imaging quality due to drift (optical astigmatism due to drift, how large is the drift and can limitation be solved?) as well as the outlook for 3 D correlation to achieve an accuracy that would be sufficient for targeted FIB-milling (~500 nm, the need for deconvolution to improve the z-resolution can this indeed be improved, and to what accuracy?)It would strengthen the paper considerably, if additional discussion/argumentation was giving detailing the possibility of a more accurate performance on next-generation systems.

The effect of astigmatism due to thermal drift, as described by the reviewer can become a major limitation and can result in a loss of resolution up to 9% relative to the expected diffraction limit of the optical setup. Accordingly, we improved the mechanical design of the microscope (the drawings provided in the supplementary material are the latest and include the improvements) and therefore increased stability of the atmospheric section of the light microscope. The changes in the astigmatism over time detected previously are not measurable anymore.

We modified the Discussion section to match the current latest design, added a description of the best alignment performance of the optical path. We added the discussion of the improvements that could lead to better 3D correlation.